# Evaluation of virtual tour in an online museum: Exhibition of Architecture of the Forbidden City

Jia Li[1]*, Jin-Wei Nie[2], Jing Ye[3]

1 Ph.D. Program in Design, Chung Yuan Christian University, Taoyuan, Taiwan, 2 Department of Interior Design, School of Design, Chung Yuan Christian University, Taoyuan, Taiwan, 3 Graduate Institute of Management, Chang Gung University, Taoyuan, Taiwan

* 519061889@qq.com

## Abstract

Online virtual museum tours combine museum authority and an academic approach with the diversity and interactivity of online resources; such tours have become an essential resource for online scientific research and education. Many important museums around the world are developing this type of online service. Comprehensive evaluation of such tours is, however, urgently needed to ensure effectiveness. This paper establishes a heuristic evaluation scale based on the literature. Taking the online virtual tour of the Exhibition of Architecture of the Forbidden City as a case study, confirmatory factor analysis was then carried out to improve the scale. Interviews were conducted to discuss and analyze the research results. The developed evaluation scale has four dimensions: *authenticity*, *interaction*, *navigation*, and *learning*. The results from the case study showed, first, that the exhibition had visual authenticity, but the behavioral authenticity was insufficient; second, the exhibition was generally interactive, but this aspect could be improved by enriching the links; third, the lack of effective navigation design for the exhibit was the main factor affecting experience quality. Fourth, the exhibition was informative and supported learning, but needs further improvement to the quantity and quality of information provided. Finally, the interviews revealed that the online exhibition did not entirely support people of different ages and abilities, so it needs further improvement to be wholly inclusive.

## Introduction

In the traditional sense, a museum collects a variety of valuable objects to primarily support scientific research and social education [1]. Going beyond the traditional methods for exhibiting material, as technological innovation and transitions in institutional function allow, museum displays are becoming increasingly diversified, and there has been a development focus on interaction, with a recent focus on enhancing the relationship between human and objects (exhibits) [2]. With the development of science and technology, museums are, in turn, developing and applying a variety of new digital technologies and thus greatly expanding the

---

**Data Availability Statement:** All relevant data are within the paper and its Supporting Information files.

**Funding:** The author(s) received no specific funding for this work.

---

**Competing interests:** The authors have declared that no competing interests exist.

ways of displaying collections across time and space and creating brand-new new experiences [3]. In 2019, International Council on Monuments and Sites (ICOMOS) promulgated *London's Charter*, which considered the digitization of cultural assets (including museum galleries) and noted that the accuracy of this approach needs careful and rigorous consideration. Different digital visualization methods and results need to be evaluated to ensure understanding of the assets through interpretive materials and to obtain the best results for the museum and its visitors. Thanks to the inherent advantages of academic authority and objectivity, museum resources have become a quality resource for online academics and education; an audience-centered experience should, however, play a real and important role in the evaluation of these and related resources [4].

An online museum is also called a virtual, digital, or electronic museum [5]. Online museums are an extension of the traditional museum but are based on network technology and contain multi-dimensional works and hypermedia [6]. Online museums break the limitations of physical time and space that bind traditional museums and can constantly update previous collections, research, displays, and educational tools [7]. Online museums shake the traditional museum's display view, which is centered on objects, and the traditional display space view, which is based on buildings as containers; thus, online museums significantly increase audience autonomy.

Virtual tour is essentially a branch of virtual reality (VR) technology and has been widely used in the medical, building, and transportation industries [8]. Unlike VR, the virtual tour space environment really exists [9]. Through the use of panoramic image technology, which acquires information about the real space environment and generates a remarkably similar VR space, people who cannot visit the museum in person can engage in an immersive experience of the museum collection through a phone or computer interface [10].

The ultimate goal of such an exhibition is the audience's cognitive improvement, as well as the perceptual and intellectual recognition people gain, as an experiencing subject, after extensive interaction with the museum exhibits, media, and space [11]. A *virtual tour of an online museum* can be regarded as an extension of the online museum and as an organic combination of the *real* and the *online museums* [12]. This involves making the museum's architectural or exhibition space into a virtual tour, which can then be publicly released through the online museum. In recent years, many world-renowned museums—including the Louvre Museum in France, the Metropolitan Museum of Art in the United States, the Palace Museum in China, and the Hermitage Museum in Russia—have opened online museum virtual tours on their official websites. If we ignore the evaluation of the effects of digital technology (as applied in museums) on the general audience experience, it will not be possible to assess the full importance of the functions of digital technology, nor can it truly make up for the deficiencies of traditional museums [13]. Given the above problems, the purpose of this study is as follows.

1. To try to construct a set of user experience evaluation methods for online museum virtual tours; and

2. To evaluate, as a case study, the Exhibition of Architecture of the Forbidden City (EAFC), to further demonstrate and improve the suggested method.

## Literature review

How can online museum virtual tours be evaluated? What would be the result of such evaluations? Donghai (1988) believes that the study of the museum audience is a critical trend in the development of contemporary museology and has become one of the standards for museum modernization, sometimes even determining the fate of the museum [14].

There has been notable research on the evaluation of virtual museum tours. Based on the development of a virtual tour application for the Ispata Museum in Turkey, Bastanlar discussed user preferences for *navigation functions*, *control options*, and *information acquired* during the virtual museum tour [15]. Barbieri (2017) evaluated three critical qualities—*usability*, *entertainment*, and *learning*—of two kinds of virtual museum systems based on the development of a virtual museum system for Cetraro in Italy. However, the evaluation method was relatively simple, and the research scope was somewhat limited [16]. Kabassi et al. evaluated a virtual tour of museums in Italy by combining the VR evaluation scale developed by Sutcliffe and Gault and concluded that the three most important dimensions in virtual museum tours are *coordination of movements and performance*, *support of navigation*, *direction*, and *support of learning* [17]. However, this study used experts as the main evaluating body, ignoring the fact that the majority of virtual museum users are not experts but rather the general public. This is problematic, as some studies have shown that, in the evaluation of the application experience, the opinions of experts and general users sometimes differ [18].

There are no direct relevant research results on user experience evaluation of virtual tours in online museums. Styliani et al. analyzed the relationship between user experience and online museums and proposed the importance of "real user" experience evaluation [11]. Pagano, Roussou, et al., through user-experience research, discussed the path and communication paradigm of improving the user interaction experience in virtual museums. However, the scope of their research was based on digital interactive services in the physical environment of the museum, rather than on online services [19, 20]. Lin collected and studied the user data from online museums and summarized four design features and five design guidelines to improve user experience, but this research mainly focused on the online learning experience [21]. Based on a literature review, MacDonald (2015) established a set of scales for evaluating user experience of online museums, which included three dimensions—*visceral*, *behavioral*, and *reflective*, which correspond to *visual*, *interaction*, and *experience*—and conducted an empirical study through expert user experience [22]. Although the above studies involved the evaluation of user experience of museum virtual or online resources, they did not involve the research issues considered in this paper.

## Materials and methods

The object of this evaluation is the EAFC, which is a permanent exhibition on the official website of the Palace Museum in Beijing. The exhibition focuses on the achievements of Lei, a famous architectural designer of the Qing Dynasty, and uses texts, drawings, photos, and models as exhibits to comprehensively show the artistic appearance of imperial architecture in the Forbidden City (the link to the exhibition is: http://quanjing.artron.net/scene/gPTvX3m1LENXdkTv5UzNsDxkLU1rUNKV/zijinchengjianzhuzhan/tour.html).

Fig 1 is the screenshot of the virtual tour interface; during the tour, the interface consists of five parts: function button, movement button, point of interest (POI), artifact information, and exhibition space. Buttons with different symbols represent different functions, such as switching scenes, zooming, and viewing maps. The white tip that appears on the floor of the exhibition space is the move button, which can be clicked to switch scenes. The little blue magnifying glass symbol near the artifacts is the POI, and when you click on the POI it pops up information about the artifacts. The exhibition space is a virtual tour space based on 360˚ panoramic photography technology. During the experience, visitors can move the view or zoom in on the artifacts with the mouse.

Based on heuristic evaluation, this study constructed an evaluation scale for the online museum virtual tour experience. Developed by Jakob Nielsen, heuristic evaluation has often

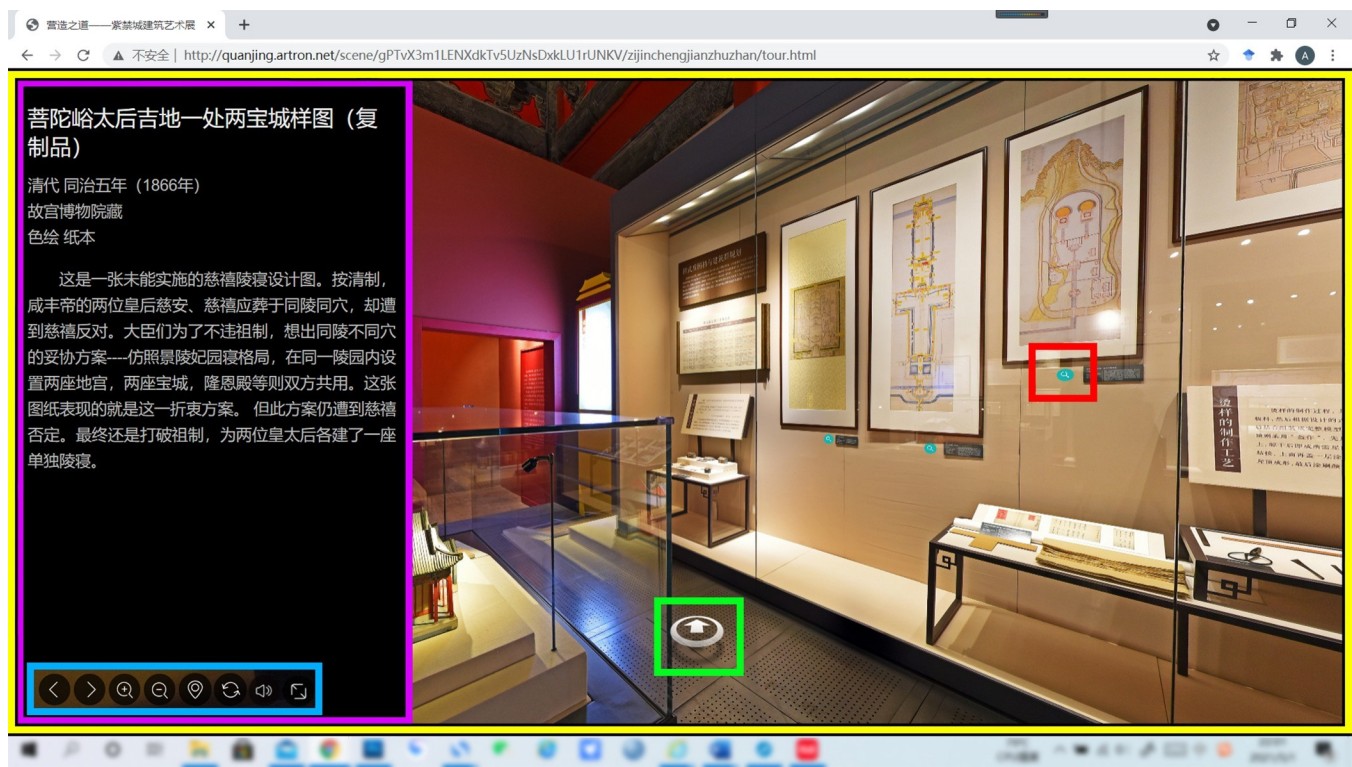

**Fig 1. The screenshot of the virtual tour interface.** (Blue, green, red, purple, and yellow boxes are drawn by the researchers, in which the blue box shows function buttons, the green box shows the movement button, the red box shows the POI, the purple box shows the artifact information, and the yellow box shows the exhibition space).

been applied to evaluating the usability of products or services. Sutcliffe and Gault (2004) developed a heuristic evaluation scale for VR with 12 factors, including *naturalness* and *compatibility* [23]. Kabassi et al. (2019) applied Sutcliffe and Gault's scale to the study of virtual tours and summarized the relevant factors into four dimensions of VR experience, presence perception, navigation ability, and learning support [17] (Table 1). In addition to revising some expressions of these dimensions and factors, the fourth dimension, *learning support*, was expanded to include four factors, such as *information availability* and *information richness*, and the 12 factors were expanded to 15, to highlight the educational function of museums (Table 2).

Confirmatory factor analysis (CFA) was the primary method used in this study and was combined with interviews. The purpose of CFA is to verify the validity of the scale and further optimize its structure.

The questionnaire was divided into two parts, A and B. Questionnaire A was designed according to the four dimensions and 15 criteria in Table 1, with a total of 18 items, each item set to strongly disagree, disagree, general, agree, and strongly agree on a five-level Likert scale. Questionnaire B was a demographic variable questionnaire and gathered information including age, gender, and educational background using a single choice format. All the collected data were analyzed using SPSS Statistics 26.

In addition to the questionnaire, the interview is also a common method used in many similar studies [24, 25], and the semi-structured interview is usually the most commonly used method for this kind of research [26–28]. The content of this interview is basically still centered on the four dimensions in the scale. The interviews sought to provide a more comprehensive and in-depth understanding of the experience of the subjects and to supplement the

**Table 1. The scale of Sutcliffe and Gault (2004) and Kabassi et al. (2019).**

| Heuristic evaluation scale for VR developed by Sutcliffe and Gault (2004) | Heuristic evaluation scale for virtual tours developed by Kabassi et al. (2019) | |
|---|---|---|
| | Factors | Categories |
| 1. Natural engagement. | 1. Natural engagement: how close the interaction is to the real world. | VR experience |
| 2. Compatibility with the user's task and domain. | 2. Compatibility with the user's task and the domain: how close the behavior of objects is to the real world and affordance for task action. | |
| 3. Natural expression of action. | 3. Realistic feedback: visibility of the effect of users' actions and conformity to the laws of physics. | |
| 4. Close coordination of action and representation. | 4. The natural expression of action: does the system allow the user to act naturally? | Perception of presence |
| 5. Realistic feedback. | 5. Close coordination of action and representation: quality of the response between user movement and virtual environment. | |
| 6. Faithful viewpoints. | 6. Clear turn-taking: clearness of who has the initiative. | |
| 7. Navigation and orientation support. | 7. Sense of presence: the naturalness of the user's perception of engagement in the system and being in a 'real' world. | |
| 8. Clear entry and exit points. | 8. Faithful viewpoints: the naturalness of change between viewpoints. | Navigation |
| 9. Consistent departures. | 9. Navigation and orientation support: naturalness in orientation and navigation. Is it clear where they are and how they return? | |
| 10. Support for learning. | 10. Clear entry and exit points: clearness of entry and exit points. | |
| 11. Clear turn-taking. | 11. Consistent departures: consistency of departure actions. | |
| 12. Sense of presence. | 12. Support for learning: promotion of learning. | learning aspect |

deficiencies of the quantitative analysis [25], so we look forward to some unexpected answers and asking further questions based on the answers.

The test process included the following steps: first, the subjects were asked to experience the EAFC online. During the experience, the subjects were asked to find a designated exhibit and then visit it freely. Second, after the tour had been completed, the subjects filled in the questionnaire to collect data for statistical analysis. Finally, sample interviews were conducted after the questionnaire had been completed. During the whole process, except in the first step, all of the participants were informed of the theme of the exhibition and were required to find a designated artifact named Ceiling of Ci Ning Palace Garden Linxi Pavilion (Fig 2); all the other artifacts were visited by the participant freely. The test time for each participant was about 10–20 minutes (Fig 3).

## Ethical statement

The study complied with the IRB principles and was approved by the Academic Committee of Jiaxing University. Before data collection took place, all participants were informed of the benefits, risks, purpose, and how the data were used. All tests were done with the participant's consent and the questionnaire and interview were completed anonymously.

## Results

To ensure the reliability of the questionnaire, this work consisted of a pre-test and an official test, and SPSS was used for reliability analysis of the data. The pre-test ran from January 4 to

**Table 2. Evaluation scale for virtual tours of online museums.**

| Dimensions | Factors | Definitions | Item |
|---|---|---|---|
| **Authenticity** | A1 Authenticity of participation | The interactive experience is as close as possible to what happens in the real world. | Q1 When I wander, I feel like I'm in a real museum. |
| | A2 Environmental authenticity | The virtual environment is as close to the real world as possible. | Q2 I felt like I was in a real museum. |
| | | | Q3 The artifacts give me a very real feeling. |
| | A3 Authenticity of feedback | The feedback of the virtual environment based on behavior corresponds to the state of the real world. | Q4 The space and objects in the virtual exhibit give real responses to my wandering behavior. |
| **Interactivity** | B1 Naturalness of behavior | Behavior is natural and unrestricted. | Q5 The process of my virtual tour is very natural and there are no restrictions. |
| | B2 Viewer-system coordination | The behavior is coordinated with the performance of the system. | Q6 When I move the camera, the picture changes very naturally. |
| | | | Q7 When I zoom out or zoom in, the picture changes very naturally. |
| | B3 Clear permissions | In the process of interaction, the rights of the viewer and the system are very clear. | Q8 I understand what I can operate and what I can't. |
| | B4 Naturalness of being | The sense of presence and participation is very natural. | Q9 When I interacted with the exhibits, the feedback was as expected. |
| **Navigation** | C1 Loyalty of perspective | The change in perspective direction is expected. | Q10 My perspective changed in line with my expectations. |
| | C2 Clarity of direction | The viewer always knows the direction. | Q11 I always knew the directions to visit. |
| | C3 Clarity of location | The viewer always knows the location. | Q12 I always know where I am. |
| | | | Q13 I know how to locate myself when I am lost |
| | C4 Clarity of start and end | The viewer knows where to start and where to end. | Q14 I know where we start and where we end. |
| **Learning** | D1 Information availability | The audience gets the information they want. | Q15 I can get the information I want. |
| | D2 Information abundance | The system can provide enough information. | Q16 I get enough information from the exhibition. |
| | D3 Enjoyable presentation of information | The information provided by the system can arouse the attention and interest of the audience. | Q17 The information I get is interesting to me. |
| | D4 Connectivity of information | The audience is willing to share their information with others. | Q18 I will discuss the information with others. |

January 13, 2020, with 22 subjects; 18 valid questionnaires were collected, and the reliability test results showed that the Cronbach's α coefficient was 0.815 ($> 0.7$). The official test ran from January 15 to February 20, 2020, and the Cronbach's α coefficient of all valid questionnaires was 0.932 ($> 0.7$). The above results prove that the scale had high reliability.

A total of 254 people participated in the official test, and 212 valid questionnaires were collected. Respondents included 108 males and 104 females, and the gender ratio for the valid questionnaires was the same as for the original sample of collected responses. Most respondents were aged 20–39 (82.54%). Most had a college degree or above (college students, 66.04%; graduate students, 26.89%). Most subjects (95.76%) were experiencing an online museum virtual tour for the first time.

'Agree' and 'strongly agree' accounted for 79.72% of all subjects' responses to the EAFC experience, and the average value for all items was 3.88, indicating that most subjects were satisfied with the experience. According to the analysis of average scores (Fig 4), *I feel I have entered a real museum* (Q2) had the highest average score (4.2), while *I always know the directions to visit* (Q11) had the lowest average score (3.6). Thus, a virtual tour based on panoramic image technology can indeed reflect the real space environment realistically. However, in the process of use, the navigation design appeared to be insufficient to affect the satisfaction of the test subjects. From the scale dimensions, the average score for *navigation* was the lowest (3.71), and the average score for *authenticity* was the highest (4.03), followed by *learning* (3.96) and

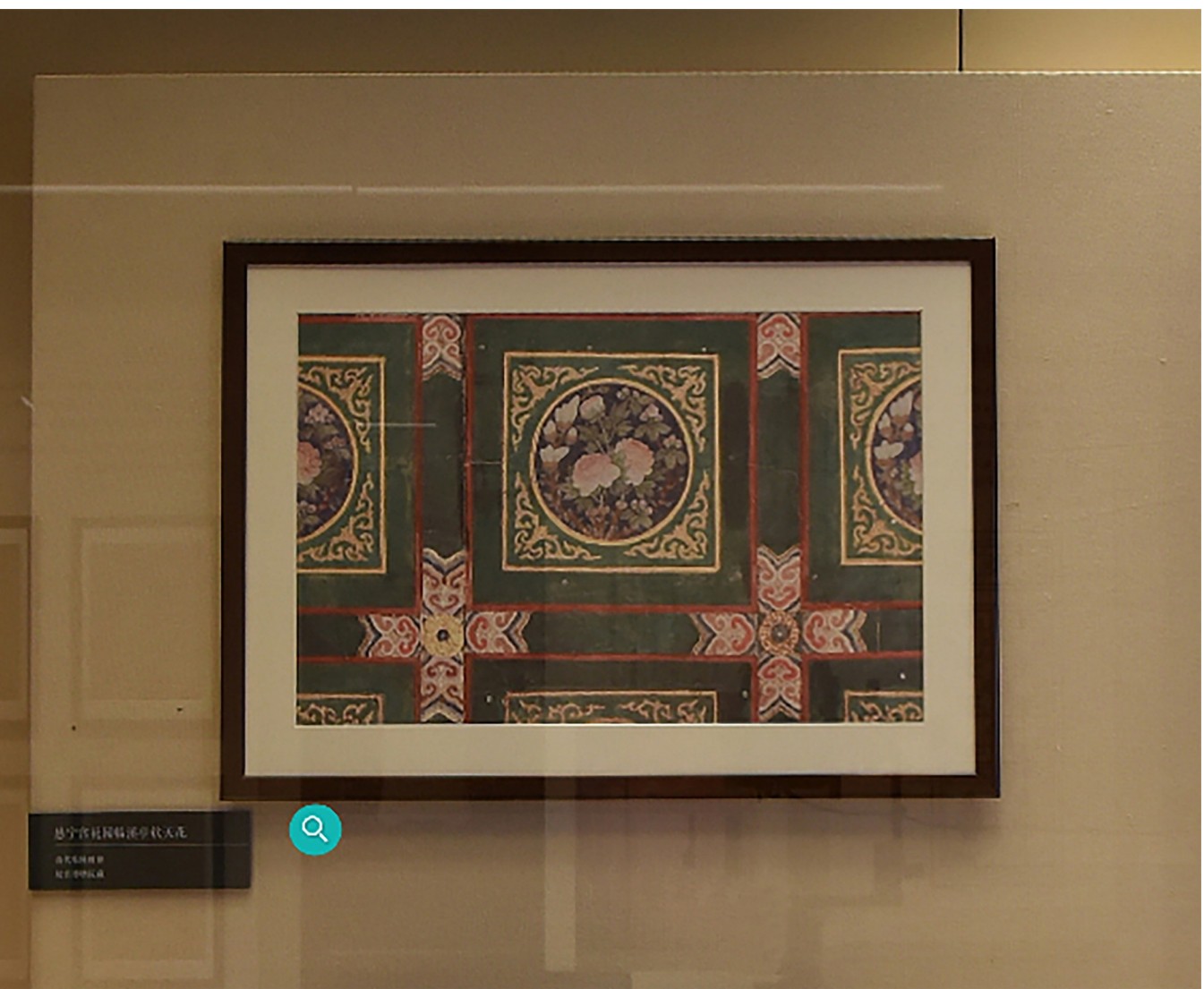

**Fig 2. Ceiling of Ci Ning Palace Garden Linxi Pavilion.**

*interactivity* (3.85). This result confirmed that the virtual tour provided a poor navigation experience, but a good experience in terms of reality and other aspects.

To judge whether the variables could be used for CFA, we conducted a Kaiser–Meyer–Olkin (KMO) test on the data and Bartlett's sphericity test, as shown in Table 3. The analysis results showed that the KMO value was 0.935, which is larger than the required 0.6 to meet the standard; the value for Bartlett's spherical Sig. is 0.000, which is less than 0.05, and this indicates that all 18 items are suitable for CFA.

Varimax was adopted to obtain the component matrix after the rotation axis, as shown in Table 4. The matrix was then rearranged according to whether the factor load coefficient was greater than 0.4 based on the corresponding relationship between the problem items. Table 3 shows that Q7 is close to the factor load coefficient of factor 1 and factor 2 (0.503; 0.474), and Q15 is close to the factor load coefficient of factor 3 and factor 4 (0.454; 0.497); we therefore tried deleting one or both of the questions. Considering the rationality of the scale structure and the component, we found that after the deletion of Q15, the corresponding relationship

| Step | Introduction | Diagram (the blue one represents the participant and the red one represents the researcher) |
|------|--------------|------------------------------------------------------------|
| 1 | The participant needs to find a designated artifact. | 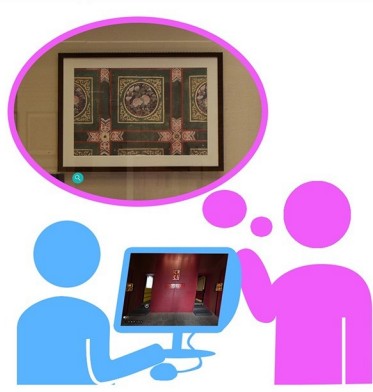 |
| 2 | The participant can visit the other exhibit freely after step 1. | 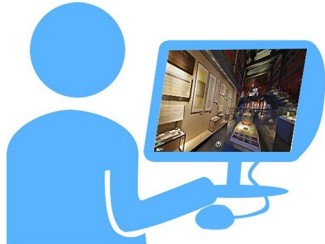 |
| 3 | After the visit, the participant fills out the questionnaire. | 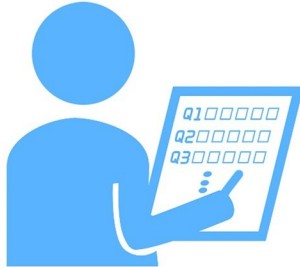 |
| 4 | After step 3, the participant was interviewed by the researcher. | 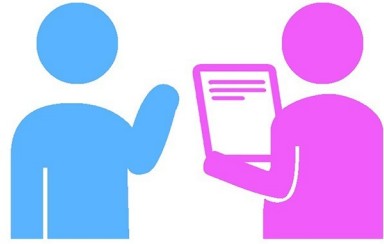 |

**Fig 3. Flow diagram of the test.**

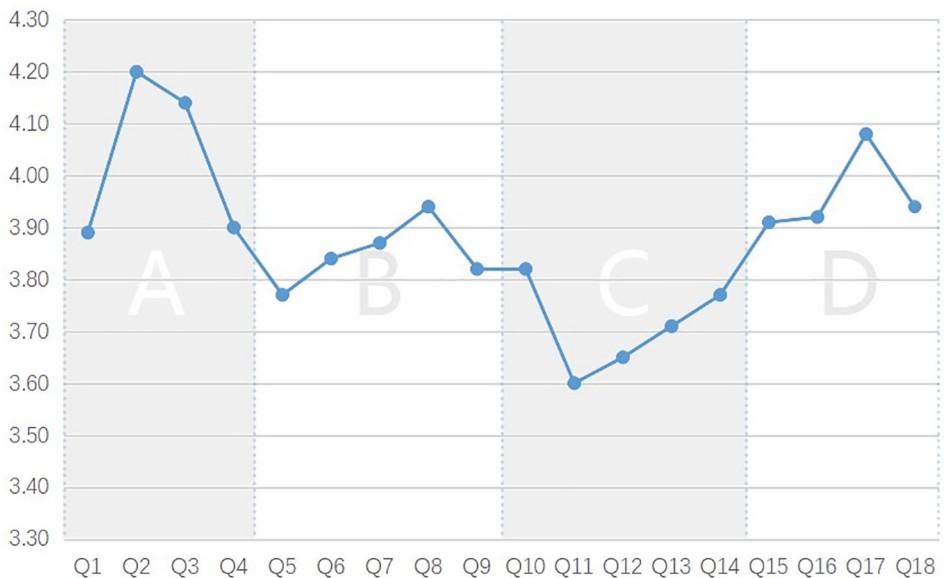

**Fig 4. Comparison of mean scores of questionnaire items.**

between the factors for each research item tended to be reasonable. The CFA results after deleting Q15 showed that Q8 was the only valid item with a component (0.661) corresponding to factor 4, and the corresponding relationship between the factors of all of the researched items tended to be reasonable after deleting attempts (Table 5).

We conducted interviews with 33 respondents, and the interviews were mainly semi-structured. We chose interviewees from different age groups, and the proportion of interviewees is as close as possible to their age structure. Due to the spread of COVID-19, we used Line or WeChat for interviews. During the interviews, the researchers made sure they were alone in the room without external interference. At the start of each interview, we requested the interviewees to be alone in a room as much as possible, and we also informed them the interview would be recorded. Fortunately, all the interviewees we contacted were cooperative. Table 6 shows the questions of the interview questionnaire, the age distribution, and the number of participants and respondents in Table 7. First, the researchers asked simple questions such as How did you feel about the experience? What were some of the problems you encountered? Second, the interviewees were asked additional questions that were designed according to the four dimensions of the scale. Finally, according to the specific responses provided by the interviewees, the interviewees were asked further questions to gain an understanding of their real experience as much as possible. Each interview was for 15–30 minutes.

The final interview results reflect the evaluation dimensions contained in the evaluation quantitative research scale mentioned above. On the whole, most of the participants expressed a positive attitude towards the experience and said their experience with the FCAA was

**Table 3. KMO and Bartlett test results.**

| Kaiser–Meyer–Olkin Measure of Sampling Adequacy | | .936 |
|---|---|---|
| Bartlett's Test of Sphericity | Approx. Chi-Square | 2070.515 |
| | df | 153 |
| | Sig. | .000 |

**Table 4. Rotated component matrix.**

| Dimensions | Items | Factor loadings | | | | Communalities |
|---|---|---|---|---|---|---|
| | | Factor1 | Factor2 | Factor3 | Factor4 | |
| A | Q1 | .806 | .166 | .138 | .199 | .736 |
| | Q2 | .771 | .197 | .161 | .256 | .724 |
| | Q3 | .702 | .124 | .171 | .342 | .654 |
| | Q4 | .642 | .098 | .272 | .304 | .588 |
| B | Q5 | .632 | .163 | .168 | .305 | .547 |
| | Q6 | .665 | .388 | .231 | -.054 | .649 |
| | Q7 | .503 | .474 | .213 | -.043 | .525 |
| | Q8 | .248 | .303 | .228 | .629 | .600 |
| | Q9 | .639 | .130 | .428 | .006 | .608 |
| C | Q10 | .694 | .340 | .285 | -.177 | .710 |
| | Q11 | .164 | .757 | .327 | -.032 | .708 |
| | Q12 | .243 | .822 | .099 | .240 | .801 |
| | Q13 | .135 | .715 | .084 | .404 | .699 |
| | Q14 | .224 | .740 | .215 | .167 | .672 |
| D | Q15 | .316 | .254 | .454 | .497 | .617 |
| | Q16 | .287 | .186 | .786 | .127 | .751 |
| | Q17 | .388 | .224 | .697 | .062 | .691 |
| | Q18 | .151 | .222 | .706 | .272 | .645 |

"realistic" or "interesting." Some respondents even said they had a sense of freshness or "surprise." Some interviewees said, however, that the tour was "inconvenient" to use, and some interviewees even said they would "feel dizzy" and have a bad experience after using it for a long time. In addition to the evaluation of the four dimensions, we found that FCAA neglected the inclusive design, which also affected the experience of some people. Therefore, we believe that inclusivity cannot be ignored. In the following section, We have combined the results of

**Table 5. Rearranged component matrix.**

| Dimensions | Items | Component | | | | Communalities |
|---|---|---|---|---|---|---|
| | | Factor1 | Factor2 | Factor3 | Factor4 | |
| A | Q3 | **.799** | .223 | .208 | .052 | .735 |
| | Q2 | **.786** | .230 | .185 | .217 | .752 |
| | Q1 | **.784** | .168 | .154 | .280 | .745 |
| | Q4 | **.660** | .133 | .291 | .209 | .582 |
| | Q5 | **.586** | .155 | .170 | .351 | .520 |
| | Q9 | **.526** | .049 | .384 | .412 | .596 |
| B | Q10 | .439 | .151 | .180 | **.696** | .732 |
| | Q7 | .282 | .318 | .187 | **.602** | .578 |
| | Q6 | .468 | .253 | .173 | **.589** | .659 |
| C | Q12 | .235 | **.837** | .130 | .214 | .819 |
| | Q13 | .230 | **.812** | .131 | .024 | .730 |
| | Q14 | .200 | **.740** | .224 | .223 | .688 |
| | Q11 | -.016 | **.628** | .271 | .513 | .731 |
| D | Q18 | .211 | .274 | **.782** | .013 | .731 |
| | Q16 | .236 | .150 | **.765** | .275 | .740 |
| | Q17 | .329 | .183 | **.685** | .275 | .687 |

**Table 6. The interview questions.**

| The overall feeling | 1) How did you feel about the experience? |
|---|---|
| | 2) What were some of the problems you encountered? |
| Authenticity | 3) Do you feel real? |
| | 4) Does it feel like visiting a real museum exhibition? |
| Interactivity | 5) How do you feel about interacting with the virtual exhibition? |
| | 6) How do you feel the virtual exhibition respond to you? |
| Navigation | 7) Do you have any trouble finding directions or exits? |
| | 8) Do you know where you are in the exhibition? |
| Learning | 9) Do you think you can learn anything from this exhibition? |
| | 10) Did you find the exhibition interesting? |

CFA and the interview to discuss our research results from the four dimensions of the scale and inclusivity.

## Discussion

According to CFA, the four dimensions of the original design were verified, and the scale structure was concise and reasonable. Through CFA, the factors *clear permissions* and *information availability*—corresponding to Q8 and Q15—were deleted, and the criteria *naturalness of behavior* and *naturalness of existence*—corresponding to Q5 and Q9—were adjusted to dimension A. The revised scale obtained is presented in Table 8.

In many ways, the quantitative analysis results—together with the interview results and their mutual confirmation—also show the omissions of quantitative research. According to the CFA and the interviewee feedback, the EAFC probably has the following significant aspects that need further attention.

### 1. Authenticity

Most of the experiencers rated *authenticity* well. The researchers believe this is due to the technical features of online virtual tour. However, in the authenticity dimension of the scale, there is a certain gap between the scores for the corresponding criteria such as *authenticity of participation*, *authenticity of feedback*, and *naturalness of participation*, and the scores for the questions of *authenticity of the environment*, which is consistent with the interview results. For example, when asked in an interview, "Does participation feel real?" the responses tended to be "not too bad" or "mostly it looks real," or statements in a similar vein.

Unlike a VR experience based on a completely virtual environment, an online museum virtual tour involves panoramic image acquisition of the real environment of the exhibition site. The exhibits and environment that experiencers see are almost the same as what could be seen at the site, so (ideally) viewers should feel similar to being on the museum site. Viewers

**Table 7. Age range and proportion of participants and respondents.**

| The age range of participants | The number of participants | The number of interviewee |
|---|---|---|
| ≤19 | 8(3.76%) | 0(0.00%) |
| 20–29 | 67(31.46%) | 12(36.4%) |
| 30–39 | 109(51.17%) | 17(51.5%) |
| 40–49 | 17(7.89%) | 2(6.06%) |
| ≥50 | 12(5.63%) | 2(6.06%) |

**Table 8. Evaluation scale after CFA.**

| Dimensions | Definition | Factors | Definition |
|---|---|---|---|
| **A. Authenticity** | Whether the feeling in the virtual space is close to the reality. | A1 Environmental authenticity | The virtual environment is as close to the real world as possible. |
| | | A2 Authenticity of participation | The interactive experience is as close as possible to what happens in the real world. |
| | | A3 Authenticity of feedback | The feedback of the virtual environment based on behavior corresponds to the state of the real world. |
| | | A4 Naturalness of behavior | Behavior is natural and unrestricted. |
| | | A5 Naturalness of being | The sense of presence and participation is very natural. |
| **B. Interactivity** | The reacts of virtual space to human active behavior. | B1 Loyalty of the perspective | The change in perspective direction is expected. |
| | | B2 Viewer-system coordination | The behavior is coordinated with the performance of the system. |
| **C. Navigation** | Identify location and direction in the virtual space. | C1 Clarity of location | The viewer always knows the location. |
| | | C2 Clarity of start and end | The viewer knows where to start and where to end. |
| | | C3 Clarity of direction | The viewer always knows the direction. |
| **D. Learning** | Learn new information during virtual tour. | D1 Connectivity of information | The audience is willing to share their information with others. |
| | | D2 Information abundance | The system can provide enough information. |
| | | D3 Enjoyable presentation of information | The information provided by the system can arouse the attention and interest of the audience. |

simulate a visit in a virtual environment, essentially switching between seamless panoramic photos. Because these panoramic photos have a high number of pixels, the picture has a certain sharpness and good visual effect, no matter whether users zoom in, zoom out, or rotate the angle of view. However, panoramic photographs do not change their two-dimensional nature and thus do not allow the viewer to engage in the same kind of interaction as a live visit.

## 2. Interactivity

*Interactivity* was rated moderately. The interactivity score was slightly lower than the average score for all questions, which indicates that the exhibition could be further improved in terms of interactivity. During the interviews, some interviewees said that they lacked the opportunity for more interaction with the exhibits when viewing the exhibition. For example, some interviewees said that they hoped to enjoy the exhibition from different angles and magnify more details. This suggests that visitors want more interaction with the exhibition.

The EAFC online virtual tour is still at the development level and lacks a diversified interactive design. Many virtual exhibitions have a link button that allows users to click to learn more information, ensuring a certain amount of interaction. However, to ensure a smoother online experience, the curator may have specific control over the size of the data, so some details cannot be sufficiently enlarged, and the linked information tends to be small-size pictures and texts. The curator should consider making up for this defect in subsequent improvements, such as providing more precise details in the links, interactive virtual exhibition models, or even online mini-games related to the exhibition, to provide visitors with a richer and more diversified interactive experience.

## 3. Navigation

The *navigation* evaluation was the lowest among all dimensions, and this significantly impacts the experience. In terms of navigability, *clarity of direction* was rated lowest, meaning that the audience had difficulty deciding what to see and what not to see during the virtual tour. Other navigational aspects, such as the audience's perception of their location and whether they had an exact starting and stopping position, were lower than the average score for all of the questions (3.88).

There are several reasons for this result. First, in the process of switching exhibition areas, the perspective after switching was not consistent with expectations, which is one reason for the poor navigation. When the scene of the exhibition area is switched during the virtual tour, the screen does not change according to the direction of the guide arrow before the switch but changes the perspective and faces the cabinet. Although the design is convenient in allowing the audience to enter the perspective of viewing the exhibits quickly, the inconsistency between the perspective and the behavior expectation during the switching process may create confusion. Second, the two guide arrows in the direction before and after the tour have the same appearance, and there is no indication or distinction of the exhibition sequence, which makes it easy for viewers to be confused. Finally, because the direction of the perspective is quite sensitive, it is easy to distort the audience's sense of time and space, leading to the loss of spatial orientation.

## 4. Learning

From the scale scores and interview responses, there appears to still be room for improvement in learning ability. In the questionnaire, the evaluation of *learning* is second only to authenticity. Most interviewees also said that the exhibits in the EAFC were quite rich, and they could get in touch with much information that they had not previously learned. However, others said that only part of the physical exhibits provided sufficient detailed written descriptions; most items lacked such descriptions. Most interviewees said that, after experiencing the online virtual tour, they would be more willing to visit the site, which indicates that the virtual tour can effectively expand the museum audience. However, it also reflects that the information provided by the virtual tour does not satisfy the curiosity of the interviewees. Therefore, although the online virtual tour of the museum provides good opportunities for learning, it still fails to reach the ideal state. Some interviewees also pointed out that the exhibit descriptions were too academic, which affected the learning effect.

## 5. Inclusivity

Inclusive details of the design were not considered sufficient, and these affect different types of users. The inclusive evaluation dimension should thus be taken into consideration. Inclusive design is a strategy to deal with globalization that considers user differences, both physical and mental [29, 30]. Of those surveyed, 5.7% were over the age of 50, and only two respondents were over 60. In the interview, some senior interviewees said that, because it was the first time they had come into contact with such things, they needed more time to get used to the virtual so that they could visit smoothly. One interviewee said the process of viewing the exhibition was "laborious" because the pictures or text provided by the exhibition were "not very clear." This result was not included in the previous scale and has been ignored by researchers.

Through further communication, we learned that some interviewees did not realize it was possible to zoom in and out of the exhibition picture at the beginning. Some interviewees failed to notice the extension function for the exhibit information or the function button in the lower-left corner of the page. Some interviewees did not even notice the existence of the guide arrow in the first place. This was true or almost all age groups, not just older people. Other interviewees said that voice functions, similar to voice guides, should be designed to help people with poor vision.

As online resources, online virtual museum tours should take into account audiences of different ages, abilities, and cultural backgrounds. This study suggests that *teaching guidance* could be set up on the initial webpage as a guide to the unfamiliar experience, which would allow users to become familiar with various functions smoothly. The visual recognizability of

the interface font and symbols could be further optimized, and sound, voice or music prompts —as well as other multi-sensory stimulus factors—could be integrated into the prompts, guidance and explanations to facilitate access for people of different ages, experiences, and abilities.

Further, although our participants did not include the disabled, it should be considered in the future, which is one of the goals of the inclusive design. In fact, great strides have been made to extend museum services to people with disabilities. For example, Cachia criticized the ocularcentrism in museum exhibition and argued that the inclusivity of exhibition could be expanded by integrating audio description into the exhibition itself and the discursive elements of the exhibition such as the catalogues, symposia, and websites [31]. The Audio Description Project (ADP) of the American Council of the Blinded (ACB) summarizes the relevant technical information and practice project of the museum for the audio description of the disabled and provides a broad reference for related research and practice [32]. In terms of virtual exhibition, Montagud et al. discussed the integration of audio description, subtitle, and sign language and other access services into the interface of museum virtual visit and compared the differences in the inclusiveness of different interfaces [33]. In addition, some studies were conducted in terms of touch and taste, in an attempt to provide more choices for the audience of museum virtual experience [34, 35]. All of these works provide more space for the disabled to accept museum virtual exhibition services.

## Conclusions

Based on a literature review, this paper constructed a set of evaluation dimensions for online museum virtual tours centered on user experiences. Based on the resulting scale, the CFA method and individual interviews were used to evaluate the EAFC online museum virtual tour. This made it possible to verify the dimensions of the scale, as well as simplifying and rationalizing the structure. Based on an evaluation of the EAFC, we drew the following conclusions:

1. The exhibition fostered a good sense of reality. Due to the characteristics of the online museum virtual tour technology, the EAFC provided good performance in terms of the authenticity of the environment and the virtual exhibit environment. However, due to the characteristics of the panoramic image acquisition technology, the EAFC provided less satisfactory visual effects in terms of the experience of the virtual environment, authenticity of human behavior feedback, and naturalness of behavior.

2. The interactivity of the exhibition is merely general and needs to be improved. The EAFC has a certain degree of interactivity, such as zooming in or out of the exhibits, and specific information can be obtained by clicking a link. However, it would possible, for example, to add a variety of different types of content to the links to make the experience more interactive.

3. The navigation design has substantial room for improvement. Navigation has a significant influence on improving the experience of virtual online museum tours. In particular, when visitors visit the EAFC, it is difficult for them to have clarity about their direction and location, which may cause confusion. This is a result of not only a technical limitation of virtual tours but also a lack of humanistic consideration in UI design, operation logic, and auxiliary design, among other factors.

4. There are opportunities for learning on the site, and it can arouse the interest of visitors, but there also appears to be a lack of learning materials. EAFC has a wealth of exhibits and materials, which are of high artistic and scientific value. Although the overall variety and

quantity of the collection are rich, the quantity and quality of the information displayed for the individual exhibits are insufficient, and the forms of learning offered are also very simple, which means this is an aspect that could be improved.

5. Inclusivity was not sufficiently considered and needs to be further optimized. This finding was mainly obtained through interviews. To achieve successful, inclusive virtual tours of an online museum, everyone should be able to visit without being limited by their ability, age, or other conditions. Therefore, to improve the quality of the visit, we need to consider the design from the perspective of inclusivity.

Unlike previous related studies, this study targeted the general public and their experiences of a virtual tour service from online museums. For the first time, the universality of such tours was considered. Previous similar studies, which focused on the experience of expert users and virtual tour services in museum exhibition halls, have practical reference value and broad application scope. For museum stuffs and exhibition designers, it is hoped that the results of this research can provide some references for consideration in the design and planning of online virtual tours of relevant museums and exhibition halls and could also be used to evaluate the relevant products or services developed. For the general audience, by encouraging people to visit this type of exhibition, people who are interested in the theme of the exhibition can indeed get new information, especially under the background of the COVID-19 pandemic, people can get a visit similar to the one in the real museum without going out of their house. However, through this work, you can understand not only the advantages of this kind of exhibition, but also the problems or difficulties that you may face, so that you can be more prepared to adapt to it better. For example, you may want to look out for any functions like a mini-map that can enhance your museum visit experience.

Follow-up studies could apply the research results to more examples to further verify the feasibility and rationality of the method presented here. The heuristic scale of this study focuses on users' evaluation of usability based on their experience. Subsequent studies should further improve the relevant evaluation methods from the non-utilitarian perspective of entertainment. Due to time and space limitations, inclusivity was not discussed in depth here; alongside an aging society and the continuous attention to social equality, inclusivity should become a vital evaluation standard for museums. Follow-up research should more deeply explore this dimension and further improve the evaluation scale.

## Supporting information

**S1 File. Data for CFA.**
(XLS)

**S2 File. Original and translation of the questionnaire.**
(DOCX)

**S3 File. Data collected in interviews.**
(DOCX)

**S4 File. Interview guideline.**
(DOCX)

**S5 File. (COREQ) 32-item checklist.**
(DOCX)

**S6 File.**
(SAV)

## Acknowledgments

We thank LetPub (www.letpub.com) for its linguistic assistance during the preparation of this manuscript.

## Author Contributions

**Data curation:** Jia Li.

**Formal analysis:** Jia Li.

**Investigation:** Jia Li.

**Methodology:** Jia Li, Jing Ye.

**Project administration:** Jia Li.

**Resources:** Jia Li.

**Software:** Jia Li, Jing Ye.

**Supervision:** Jia Li, Jin-Wei Nie.

**Validation:** Jia Li, Jin-Wei Nie, Jing Ye.

**Visualization:** Jia Li.

**Writing – original draft:** Jia Li.

**Writing – review & editing:** Jia Li, Jin-Wei Nie, Jing Ye.

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
