## [Decision Letter · Decision Letter 0]

8 Apr 2021

PONE-D-21-07531

Evaluation of Virtual Roaming in an Online Museum: Exhibition of Architecture of the Forbidden City

PLOS ONE

Dear Dr. Jia,

Thank you for submitting your manuscript to PLOS ONE. After careful consideration, we feel that it has merit but does not fully meet PLOS ONE’s publication criteria as it currently stands. Therefore, we invite you to submit a revised version of the manuscript that addresses the points raised during the review process.

Two Reviewers evaluated the manuscript and gave generally favorable opinions, however the contribution needs extensive revisions. Especially: 

- language editing by a native english speaker is still needed

- the link to the online exhibition did not work at my first try; you should probably separate it from the final dot (without it, it works)

- methodology lacks information, especially regarding the methodological approach for qualitative data

- ethical approval was missing. This is necessary for consideration by PLOS ONE

- Authors report a pre-test of the questionnaire and an alpha with 18 participants. Is this useful? What was the reliability index with the full sample?

- data were partial. First, items in the database are in chinese, they should be translated in english to be readable by international audience; second, qualitative data from interviews were missing. These are also mandatory for consideration by PLOS ONE

- figures and more information on participants' experience and tasks within the exploration would improve the readers' comprehension of the research

- final discussion says little about the take-home messages for a broader audience: what have we learnt about the design and evaluation of online museums in general? 

I encourage Authors to perfom extensive revisions taking into account these aspects and all those identified by Reviewers. Please remind that final acceptance of the contribution could not be guaranteed at this step.

We look forward to receiving your revised manuscript.

Kind regards,

Stefano Triberti, Ph.D.

Academic Editor

PLOS ONE

Journal Requirements:

4. Please amend the manuscript submission data (via Edit Submission) to include author Jia Li.

5. Please amend your authorship list in your manuscript file to include author Taiwan Jia

Additional Editor Comments (if provided):

Reviewers' comments:

Reviewer's Responses to Questions

**Comments to the Author**

1. Is the manuscript technically sound, and do the data support the conclusions?

Reviewer #1: Yes

Reviewer #2: Yes

2. Has the statistical analysis been performed appropriately and rigorously? 

Reviewer #1: I Don't Know

Reviewer #2: Yes

3. Have the authors made all data underlying the findings in their manuscript fully available?

Reviewer #1: Yes

Reviewer #2: No

4. Is the manuscript presented in an intelligible fashion and written in standard English?

Reviewer #1: Yes

Reviewer #2: Yes

5. Review Comments to the Author

Reviewer #1: This is an excellent research paper of relevance to museum studies and it should be published. The methods and process are well-described and clear. The paper is well-written. I am not a statistician so I can't evaluate this part. However, I would be happy to use the paper and evaluation tool in my own teaching in museum studies. My "minor revisions" suggestions are very minor. First, on p. 3, ICOMOS should be defined or the name written out fully. Next, on p. 9, in the chart, at Learning D3 the term "Fun of knowledge" is written. This is not correct English usage. Instead, it should say something like "Enjoyable presentation of information." Then, on p. 22, the term "generality" is used in reference to a suggestion by the authors that the concept should become a "vital evaluation standard." The term's application should be defined and an example included, in my view. Last, on p. 18-19 and again on p. 21 the authors refer to inclusivity, inclusiveness and ability. Yet, they do not mention disability (an oversight, in my view), and the solutions offered through disability studies and inclusive design, such as image descriptions, audio descriptions, and so on. For example, see: https://www.acb.org/adp/ There are many such strategies recommended for museums by disability studies scholars and activists--see Amanda Cachia and Georgina Kleege, for example. The paper should direct museum-based readers to look to these and related sources for suggestions.

Reviewer #2: The manuscript discusses a research to develop a heuristic evaluation scale based on the existing literature and a case study. The case study is a virtual online museum tour. The proposed evaluation scale includes four dimensions: authenticity, interaction, navigation, and learning.

I am not expert in factor analysis, and I assume that the methodology is properly applied.

My concerns are related to the description of the case study and the methods.

First of all, I suggest the authors to better describe the case study. In addition to the link provided to access the online tour, there is the need to understand the functionalities, how the experience is structured, the peculiarities of the museum as well as the interaction modalities. Adding screenshots of relevant points of interest can help to better understand the case study.

Secondly, some details related to the survey are missing:

- Since the ethics statement is not provided, I wonder whether the authors deal with the informed consent to involve the participants in the research.

- How was the EAFC online experience presented to the participants? I wonder whether the authors provided some instructions to the participants about the navigation and the interface, or just let them free to explore the functionalities.

- What was the task to accomplish during the experience? Was it the same for all the participants?

- What were the groups and attributes used for the stratified sampling of the interviewees?

- It is not clear the reason why the researcher asked “further random questions” during the interview. Is “random” the proper term?

- Moreover, it is not clear how the qualitative data from the interviews were analysed and integrated with the questionnaire’s results.

Finally, I suggest the authors to introduce the scale by conceptualizing the different elements, as well as to clearly explain how their scale differs from and improve the existing literature.

A final comment is about the brief discussion about inclusivity, that I appreciate. Indeed it is a relevant issue to make virtual museum accessible for people with diverse needs and abilities. I hope that the authors will further explore this aspect in details, because the literature is missing such investigations to guide the future design.

6. PLOS authors have the option to publish the peer review history of their article (what does this mean?). If published, this will include your full peer review and any attached files.

Reviewer #1: No

Reviewer #2: **Yes: **Annamaria Recupero

---

## [Author Response · Author response to Decision Letter 0]

7 Jun 2021

Response to Reviewers

Reply to reviewer’s comments: Manuscript [PONE-D-21-07531]

Dear Dr. Stefano Triberti,

We would like to thank the editor and reviewers for the invaluable comments and advice that helped improve our original manuscript. My partners and I studied the journal and reviewer's suggestions for revision carefully and revised the manuscript based on each suggestion. Please find our responses to the comments below.

Editor Requirements

1. language editing by a native english speaker is still needed.

Authors’ response: Thank you for your suggestion. We will continue to hand over the manuscript to a professional translation company for editing.

2. the link to the online exhibition did not work at my first try; you should probably separate it from the final dot (without it, it works).

Authors’ response: Thank you for your reminder. I have reedited this link to ensure its validity. 

3. Methodology lacks information, especially regarding the methodological approach for qualitative data.

Authors’ response: Thank you for pointing out this problem. It was true that there was a lack of description in the methodology of the qualitative part of this study. We have strengthened this part in line 4-10 of p. 11 and lines 1–7 of p. 12 and added relevant references.

4. Ethical approval was missing. This is necessary for consideration by PLOS ONE.

Authors’ response: Thank you very much for your reminder. This paper did not provide an ethics statement, which was an oversight error on our part. We have added the ethical statements on p. 13, lines 3–8. 

In accordance with IRB standards, the tests were carried out under the background that the participants fully understood the purpose, process, risk, and data use of the study. The participants filled in the questionnaire anonymously, and the data were recorded with their consent. This study adopts a paperless electronic questionnaire. Only with the consent of the participants, they can continue to fill in the questionnaire. 

5. Authors report a pre-test of the questionnaire and an alpha with 18 participants. Is this useful? What was the reliability index with the full sample?

Authors’ response: Thank you for your correction. It is our negligence that we did not report the Cronbach’s α coefficient of all valid questionnaires in the study. We have added this on p. 14, lines 2–7. 

6. Data were partial. First, items in the database are in chinese, they should be translated in english to be readable by international audience; second, qualitative data from interviews were missing. These are also mandatory for consideration by PLOS ONE.

Authors’ response: We apologize for this, and we will translate the database in the future. Qualitative data will also be translated, collated, and uploaded. 

7. Figures and more information on participants' experience and tasks within the exploration would improve the readers' comprehension of the research.

Authors’ response: Thank you very much for your suggestions. We have added Figure 2 and Figure 3 in p. 12¬–13 to make our tests clearer and easier to understand.

8. Final discussion says little about the take-home messages for a broader audience: what have we learnt about the design and evaluation of online museums in general?

Authors’ response: Thank you very much for your reminder. It was an oversight error that we considered more about museum staffs and designers than a broader audience. We have added about this in the conclusion on p. 28, lines 3–10, with implications for a wider audience. 

Journal Requirements

Authors’ response: Thank you very much for your reminder. We have modified our manuscript according to the requirements of the journal to conform to PLOS ONE's style. 

Authors’ response: Thank you for your reminding. I have registered the ORCID ID and updated my personal information. 

Authors’ response: We apologize that it was an oversight error on our part to not upload these questionnaires. We have uploaded the original and translated version of our questionnaire in the Support Information.

4. Please amend the manuscript submission data (via Edit Submission) to include author Jia Li. 

Authors’ response: Thank you for the reminder. This was an editing error; we have added Jia Li in the edited submission. 

5. Please amend your authorship list in your manuscript file to include author Taiwan Jia. 

Authors’ response: Thank you for the reminder; “Taiwan Jia” was a mistake in editing, and there is no such author. We have modified the personal information again.

6. Please include captions for your Supporting Information files at the end of your manuscript, and update any in-text citations to match accordingly.

Authors’ response: Thank you for reminding us. We have added Supporting Information on p. 32 and updated our in-text citations.

Reviewer #1

This is an excellent research paper of relevance to museum studies and it should be published. The methods and process are well-described and clear. The paper is well-written. I am not a statistician so I can't evaluate this part. However, I would be happy to use the paper and evaluation tool in my own teaching in museum studies. My "minor revisions" suggestions are very minor. First, on p. 3, ICOMOS should be defined or the name written out fully. Next, on p. 9, in the chart, at Learning D3 the term "Fun of knowledge" is written. This is not correct English usage. Instead, it should say something like "Enjoyable presentation of information." Then, on p. 22, the term "generality" is used in reference to a suggestion by the authors that the concept should become a "vital evaluation standard." The term's application should be defined and an example included, in my view. Last, on p. 18-19 and again on p. 21 the authors refer to inclusivity, inclusiveness and ability. Yet, they do not mention disability (an oversight, in my view), and the solutions offered through disability studies and inclusive design, such as image descriptions, audio descriptions, and so on. For example, see: https://www.acb.org/adp/ There are many such strategies recommended for museums by disability studies scholars and activists--see Amanda Cachia and Georgina Kleege, for example. The paper should direct museum-based readers to look to these and related sources for suggestions. 

Authors’ response: Thank you very much for your recognition and support, which have encouraged us to continue improving our manuscript. In light of your suggestions, we have made the following revisions.

We added the full name for the ICOMS on p. 3, line 11.

We replaced "Fun of information" in D3 in Table 2 (original Table 1) on p. 11 with “Enjoyable presentation of information”. 

“Generality” in p. 28, lines 16–17 (original p. 22) was a wrong word, while the correct one should be “inclusivity”. The explanation of inclusivity has been defined on p. 24, line 8-9. 

Thank you very much for your suggestions on inclusion; we have learnt a lot from the links you provided. We have discussed this more on p. 25, lines 10–23 and p. 26, lines1–2, with references to some relevant literature.

Thank you very much for pointing out the mistakes in out manuscript and putting forward these valuable suggestions, which made our work more reasonable and easier to understand.

Reviewer #2

1. The manuscript discusses a research to develop a heuristic evaluation scale based on the existing literature and a case study. The case study is a virtual online museum tour. The proposed evaluation scale includes four dimensions: authenticity, interaction, navigation, and learning.

I am not expert in factor analysis, and I assume that the methodology is properly applied.

My concerns are related to the description of the case study and the methods.

Authors’ response: Thank you very much for your recognition and support.

2. First of all, I suggest the authors to better describe the case study. In addition to the link provided to access the online tour, there is the need to understand the functionalities, how the experience is structured, the peculiarities of the museum as well as the interaction modalities. Adding screenshots of relevant points of interest can help to better understand the case study.

Authors’ response: Thank you very much for your suggestions. On p. 7, lines 10–19, we described the functionalities, structure, peculiarities, interaction modalities and other elements of the exhibition. We also illustrated them through Figure 1on p. 8.

3. Secondly, some details related to the survey are missing:

-Since the ethics statement is not provided, I wonder whether the authors deal with the informed consent to involve the participants in the research.

Authors’ response: Thank you for pointing this out, and we did forget to provide an ethics statement. We have added relevant statements on p. 13, lines 3–8. The participants have fully understood the purpose, process, risk and data use of the study. They filled in the questionnaire anonymously, and the data were recorded with their consent. This study adopted a paperless electronic questionnaire. The questionnaires were filled only with the consent of the participants.

- How was the EAFC online experience presented to the participants? I wonder whether the authors provided some instructions to the participants about the navigation and the interface, or just let them free to explore the functionalities.

Authors’ response: Thank you for raising this point, and we have added description on p. 12–13 of the paper and Figure 2 and Figure 3 to further explain the testing process. Participants were only asked to find the designated artifact at the beginning; the rest of their visit was undisturbed. 

- What was the task to accomplish during the experience? Was it the same for all the participants?

Authors’ response: Yes, all the participants had the same task. They all needed to find the artifact named Ceiling of Ci Ning Palace Garden Linxi Pavilion. I apologize for that we forgot to explain in the article. We have added the description and a figure on p. 12, lines 12–17.

- What were the groups and attributes used for the stratified sampling of the interviewees?

Authors’ response: Thank you for pointing out this missing part. We did lack description of the groups and attributes used for the stratified sampling of the interviewees. However, we have strengthened the description on p. 18, lines 4–8 and added Table 7 on p. 18.

- It is not clear the reason why the researcher asked “further random questions” during the interview. Is “random” the proper term?

Authors’ response: Thank you for your correction. We also think that “random” is not very appropriate. We have re-described the interview questioning process on p. 17, line12-13 and p.18 1–2. 

- Moreover, it is not clear how the qualitative data from the interviews were analysed and integrated with the questionnaire’s results. 

Authors’ response: Thank you for pointing out our oversight error. The questions of our semi-structured interview were formulated based on the scale dimensions of quantitative research, and the discussion was also based on the results of these two parts. We gave explanations on p. 17, line 12-13 and p. 18, lines1–2, and added Table 6 on p. 18 for further explanation. 

4. Finally, I suggest the authors to introduce the scale by conceptualizing the different elements, as well as to clearly explain how their scale differs from and improve the existing literature.

Authors’ response: Thanks for your suggestion; we have conceptualized the different elements in the scale in Table 8 on p. 19–20. In addition, we have listed the scales we refer to in Table 1 on p. 9 to provide the reader with a comparison for the scales we have designed.

5. A final comment is about the brief discussion about inclusivity, that I appreciate. Indeed it is a relevant issue to make virtual museum accessible for people with diverse needs and abilities. I hope that the authors will further explore this aspect in details, because the literature is missing such investigations to guide the future design. 

Authors’ response: Thank you very much for your recognition. On p. 25, lines 10–23 and p. 26, lines 1–2 we have discussed more about the dimension of inclusivity and have cited some relevant literatures, hoping to provide more information to readers.

---

## [Decision Letter · Decision Letter 1]

5 Jul 2021

PONE-D-21-07531R1

Evaluation of Virtual Roaming in an Online Museum: Exhibition of Architecture of the Forbidden City

PLOS ONE

Dear Dr. li,

Thank you for submitting your manuscript to PLOS ONE. After careful consideration, we feel that it has merit but does not fully meet PLOS ONE’s publication criteria as it currently stands. Therefore, we invite you to submit a revised version of the manuscript that addresses the points raised during the review process.

Revisions were adequately performed by Authors, but lacks regarding the journal's criteria for publication stay unsolved

- the items in the .sav file are still in chinese. Authors are asked to "maximize the accessibility and reusability of the data", so as I said in the previous round of revisions these should be translated in the english language like the rest of the paper materials https://journals.plos.org/plosone/s/data-availability 

- the methodological details added on the qualitative research are limited. According to Author guidelines, "Qualitative research studies should be reported in accordance to the Consolidated criteria for reporting qualitative research (COREQ) checklist or Standards for reporting qualitative research (SRQR) checklist" see this page for links to the checklists https://journals.plos.org/plosone/s/submission-guidelines , I also suggest Authors to consider other qualitative research published on PLOS ONE to see examples of the checklists included as supporting information, such as for example https://journals.plos.org/plosone/article?id=10.1371/journal.pone.0247121 , https://journals.plos.org/plosone/article?id=10.1371/journal.pone.0225534

Again, please notice these are PLOS ONE's criteria for publication so they should be satisfied before the article could be considered for publication

We look forward to receiving your revised manuscript.

Kind regards,

Stefano Triberti, Ph.D.

Academic Editor

PLOS ONE

Journal Requirements:

Reviewers' comments:

Reviewer's Responses to Questions

**Comments to the Author**

1. If the authors have adequately addressed your comments raised in a previous round of review and you feel that this manuscript is now acceptable for publication, you may indicate that here to bypass the “Comments to the Author” section, enter your conflict of interest statement in the “Confidential to Editor” section, and submit your "Accept" recommendation.

Reviewer #2: All comments have been addressed

2. Is the manuscript technically sound, and do the data support the conclusions?

Reviewer #2: Yes

3. Has the statistical analysis been performed appropriately and rigorously? 

Reviewer #2: I Don't Know

4. Have the authors made all data underlying the findings in their manuscript fully available?

Reviewer #2: Yes

5. Is the manuscript presented in an intelligible fashion and written in standard English?

Reviewer #2: Yes

6. Review Comments to the Author

Reviewer #2: I appreciate the adjustments made to clarify some issues.

I found one typo on page 16 line 6: "Frist" instead of "First".

7. PLOS authors have the option to publish the peer review history of their article (what does this mean?). If published, this will include your full peer review and any attached files.

Reviewer #2: **Yes: **Annamaria Recupero

---

## [Author Response · Author response to Decision Letter 1]

14 Aug 2021

Editor Requirements

1.The items in the .sav file are still in Chinese. Authors are asked to "maximize the accessibility and reusability of the data", so as I said in the previous round of revisions these should be translated in the English language like the rest of the paper materials https://journals.plos.org/plosone/s/data-availability

Authors’ response: Thank you very much for pointing this out, and we apologize for this negligence. We have translated all the contents of the .sav file into English.

2.The methodological details added on the qualitative research are limited. According to Author guidelines, "Qualitative research studies should be reported in accordance to the Consolidated criteria for reporting qualitative research (COREQ) checklist or Standards for reporting qualitative research (SRQR) checklist".

Authors’ response: Thank you for reminding us. According to the COREQ checklist, we have supplemented more details of the qualitative study in line 9 on p. 11, line 7 on p. 13, lines 6–11 on p. 16, and line 6 on p. 17. Moreover, files S4 and S5 have been added to provide supporting information.

3.If applicable, we recommend that you deposit your laboratory protocols in protocols.io to enhance the reproducibility of your results.

Authors’ response: Thank you for your suggestion. We have uploaded the lab protocols to protocols.io, and you can view them at dx.doi.org/10.17504/protocols.io.bww4pfgw.

Journal Requirements

Please review your reference list to ensure that it is complete and correct. If you have cited papers that have been retracted, please include the rationale for doing so in the manuscript text, or remove these references and replace them with relevant current references. Any changes to the reference list should be mentioned in the rebuttal letter that accompanies your revised manuscript. If you need to cite a retracted article, indicate the article s retracted status in the References list and also include a citation and full reference for the retraction notice.

Authors’ response: Thank you very much for the reminder. We went through the references one by one and found that some of the non-English references were not easy for global readers to search and access on Google Scholar, so we replaced them with English sources with the same views. The references are modified in lines 10–21 on p. 28, lines 7–9, 13–17, and 22–25 on p. 29, and in line 1 on p. 30. 

Some new modifications are also listed below. We replace the virtual roaming with virtual tour in the manuscript. Although our research object is called virtual tour in some literatures (e.g., ref. 8), more literatures use virtual tour (e.g., ref. 14 and 16). After careful investigation, we decided to use virtual tour to clarify our research object. These are modified in the title page, line 6, 8 on p. 4, line 10–11, 17, 20 on p.7, table 2, table 8, and line 3 on p. 20.

Reviewer #2

I appreciate the adjustments made to clarify some issues. I found one typo on page 16 line 6: "Frist" instead of "First".

Authors’ response: Thank you very much for your approval, as well as for pointing out our flaws. We have corrected the misspellings in line 13 on p. 16.

---

## [Decision Letter · Decision Letter 2]

7 Dec 2021

Evaluation of Virtual Tour in an Online Museum: Exhibition of Architecture of the Forbidden City

PONE-D-21-07531R2

Dear Dr. li,

We’re pleased to inform you that your manuscript has been judged scientifically suitable for publication and will be formally accepted for publication once it meets all outstanding technical requirements.

Kind regards,

Prabhat Mittal, Ph.D.

Academic Editor

PLOS ONE

Reviewers' comments:

Reviewer's Responses to Questions

**Comments to the Author**

1. If the authors have adequately addressed your comments raised in a previous round of review and you feel that this manuscript is now acceptable for publication, you may indicate that here to bypass the “Comments to the Author” section, enter your conflict of interest statement in the “Confidential to Editor” section, and submit your "Accept" recommendation.

Reviewer #2: All comments have been addressed

Reviewer #3: All comments have been addressed

Reviewer #4: All comments have been addressed

2. Is the manuscript technically sound, and do the data support the conclusions?

Reviewer #2: Yes

Reviewer #3: Yes

Reviewer #4: Partly

3. Has the statistical analysis been performed appropriately and rigorously? 

Reviewer #2: I Don't Know

Reviewer #3: Yes

Reviewer #4: Yes

4. Have the authors made all data underlying the findings in their manuscript fully available?

Reviewer #2: Yes

Reviewer #3: Yes

Reviewer #4: Yes

5. Is the manuscript presented in an intelligible fashion and written in standard English?

Reviewer #2: Yes

Reviewer #3: Yes

Reviewer #4: Yes

6. Review Comments to the Author

Reviewer #2: (No Response)

Reviewer #3: The authors of the paper has responded and added corrections to the majority of the suggestions highlighted by the previous reviewers. It was a an acceptable paper with justifiable methodology. I would suggest additional information to be added in the methodology section (a few of the suggestions stated below were mentioned in the later part of the discussion section):

1-To clearly state the type of interview carried out (focus group or personal interview)

2-To state the selection process of the respondents and how the questionnaires were distributed (it was mentioned in the flow diagram in Figure 3, however this need to be explained earlier in the method section)

3-The sentence "..212 valid questionnaires were collected." It should be written as "...212 valid responses from the questionnaire were collected."

4-In of of the result's section, the authors mentioned that they conducted the interview by using WeChat (due to the COVID-19 pandemic). The authors need to explain the process of acquiring the contact information of the potential respondents in the methodology section.

Reviewer #4: In this paper factor analysis is conducted to obtain a clear pattern of loading. It is a way of carrying out a particular task that is used to reduce a large number of variables into fewer numbers of factors. This technique extracts maximum common variance from all variables and puts them into a common score. The loading purpose indicates the depth of the relationships between items. Each factor will tend to have either large or small loading's of any particular variable. Hence, factor loading is used to assess the validity of an item and to summarize the sort of correlation among variables. Unfortunately, this paper applied the first part but neglected the second part which is related to summarize the sort of correlation among variables. Hence, we think it needs to go further analysis like Correlation analysis to explore the relationships among the variables. The role of correlation is to capture the similarities or differences between the variables. It measures the degree of association between the values of related variables given in the data set. Then, the mutual influence of variables on one another will be traced.

It is interesting to note, that the main goal of factor analysis is to identify a group of inter-related variables, to see how they are related to each other or we can say that Factor analysis can be used to identify the hidden dimensions or constructs which may or may not be apparent from direct analysis.

7. PLOS authors have the option to publish the peer review history of their article (what does this mean?). If published, this will include your full peer review and any attached files.

Reviewer #2: No

Reviewer #3: No

Reviewer #4: **Yes: **Dr.Salahaddin Yasin Baper

---

## [Editor Report · Acceptance letter]

20 Dec 2021

PONE-D-21-07531R2 

Evaluation of Virtual Tour in an Online Museum: Exhibition of Architecture of the Forbidden City 

Dear Dr. Li:

I'm pleased to inform you that your manuscript has been deemed suitable for publication in PLOS ONE. Congratulations! Your manuscript is now with our production department. 

Kind regards, 

on behalf of

Dr. Prabhat Mittal 

Academic Editor

PLOS ONE